# Applying a genetic risk score model to enhance prediction of future multiple sclerosis diagnosis at first presentation with optic neuritis

Pavel Loginovic [1,17], Feiyi Wang [2,17], Jiang Li [3,17], Lauric Ferrat [4], Uyenlinh L. Mirshahi [3], H. Shanker Rao [3], Axel Petzold [5,6,7], Jessica Tyrrell [8], Harry D. Green [9], Michael N. Weedon[4], Andrea Ganna[2,10], Tiinamaija Tuomi [2,11,12,13], David J. Carey[3], UKBB Eye & Vision Consortium*, FinnGen*, Geisinger-Regeneron DiscovEHR Collaboration*, Richard A. Oram [4,14,18] ✉ & Tasanee Braithwaite [15,16,18]

Optic neuritis (ON) is associated with numerous immune-mediated inflammatory diseases, but 50% patients are ultimately diagnosed with multiple sclerosis (MS). Differentiating MS-ON from non-MS-ON acutely is challenging but important; non-MS ON often requires urgent immunosuppression to preserve vision. Using data from the United Kingdom Biobank we showed that combining an MS-genetic risk score (GRS) with demographic risk factors (age, sex) significantly improved MS prediction in undifferentiated ON; one standard deviation of MS-GRS increased the Hazard of MS 1.3-fold (95% confidence interval 1.07–1.55, $P < 0.01$). Participants stratified into quartiles of predicted risk developed incident MS at rates varying from 4% (95%CI 0.5–7%, lowest risk quartile) to 41% (95%CI 33–49%, highest risk quartile). The model replicated across two cohorts (Geisinger, USA, and FinnGen, Finland). This study indicates that a combined model might enhance individual MS risk stratification, paving the way for precision-based ON treatment and earlier MS disease-modifying therapy.

Optic neuritis (ON) presents most frequently in young adults with subacute uni- or bilateral vision loss[1]. It is a rare but treatable cause of blindness. The incidence of ON has been stable over decades, and varies by latitude, with a population-based incidence of 3.7 to 5.1 per 100k person-years in the United Kingdom (UK) and United States of America (USA), respectively[2,3]. Approximately two-thirds are undifferentiated at presentation with the remainder having either a prior diagnosis of Multiple Sclerosis (MS) or preceding infectious or immune-mediated inflammatory disease (I-IMID)[2]. By five years of follow-up, approximately 20% of undifferentiated ON cases are diagnosed with MS, compared to 0.1% controls (adjusted Hazard Ratio [aHR] 285, $P < 0.001$)[2]. By 15 years, up to 50% of all ON cases, excluding those with bilateral presentation, are diagnosed with MS[4,5].

Importantly, ON ultimately associated with diagnosis of MS (MS-ON)[6], including Clinically Isolated Syndrome (CIS, consisting of ON plus magnetic resonance imaging features of demyelination at presentation)[1], has different management and prognosis to non-MS-associated ON. In MS-ON, vision usually recovers spontaneously to near-baseline over 3 months[7]. Trial evidence indicates an equivocal role for corticosteroid therapy[8–10], although there may be a role for

A full list of affiliations appears at the end of the paper. *Lists of authors and their affiliations appear at the end of the paper. ✉ e-mail: R.Oram@exeter.ac.uk

hyperacute corticosteroid therapy[11]. Non-MS ON may be associated with subsequent diagnosis of corticosteroid-responsive diseases including sarcoidosis, neuromyelitis optica spectrum disorder (NMOSD), and vasculitides[12]. In marked contrast to MS-ON, axonal injury can be swift and vision loss irreversible, with significant impacts on patients' lives[13]. Clinicians managing acute undifferentiated ON face a challenging and time-critical decision while awaiting diagnostic investigations[14]: whether or not to initiate potentially sight-saving corticosteroid therapy which risks serious adverse effects[15]. There is an unmet clinical need for a tool to improve acute risk stratification, differentiating those at low future MS risk, who may benefit from urgent corticosteroids, from those at high future MS risk, who may benefit from earlier disease-modifying therapy to reduce long-term neurological disability[16,17].

Many autoimmune and autoinflammatory diseases, including MS, are heritable. Almost 20% of MS risk heritability can be attributed to common genetic variants[16,18], and the latest genome-wide association studies (GWAS) from the International Multiple Sclerosis Genetics Consortium (IMSGC)[19] study of 47,429 MS patients and 68,374 control subjects identified over 200 associated loci[20]. The identification of strong, complex, human leukocyte antigen (HLA) class II associations, combined with non-HLA associations, offers the opportunity to aggregate MS genetic risk as a continuous MS genetic risk score (GRS). Additional risk factors for MS include female sex, age at onset[21], latitude of country of residence, low serum 25-hydroxyvitamin[22–24], increased body mass index[22,24–26], Epstein-Barr virus seropositivity[27], and smoking[28,29]. In the first MS-GRS model, De Jager et al. (2009) studied 2215 individuals with MS and 2189 controls, and in independent samples confirmed that 16 MS susceptibility alleles (2 MHC alleles and 14 non-MHC alleles) had modest discriminatory ability, which was enhanced by integrating of non-genetic risk factors (sex (ROC AUC 0.74), smoking and anti-EBV antibody titers (ROC AUC 0.68) in the model[30]. Their study included subjects with MS-ON, including CIS, and showed that they share a similar genetic architecture as those with MS. To our knowledge, genetic data has not previously been used in combination with other risk factors to aid MS risk stratification in undifferentiated ON which includes, but is not limited to, CIS. This UK Biobank (UKBB) study had two aims: Firstly, to determine whether an MS-GRS, created using published GWAS summary statistics, aids prediction of future MS in people presenting with undifferentiated ON; secondly, whether combining MS-GRS with demographic and clinical variables enhances MS diagnosis risk stratification at first ON presentation.

## Results

### MS, ON and MS-ON cases and demographic characteristics

From 483,506 unrelated individuals with available genetic and phenotype data, of whom 83.9% were of European ancestry, we identified 2369 MS cases (prevalence 0.49%, or 490 per 100,000 participants) and 687 ON cases (prevalence 0.14% or 142 per 100,000 participants) (Fig. 1). ON cases included 545 (545 out of 687, 79.3%) who were not known to have MS at first ON presentation and 142 (20.7%) with prior or simultaneous diagnosis of MS (MS-ON). During cumulative follow-up from first ON presentation to latest data extraction in 2019 or death (median 18.4 years, IQR 9.9–30.2) a further 124 out of 545 (22.8%) were diagnosed with MS.

Demographic and clinical characteristics are summarised in Table 1 and Supplementary Table 5 for the cases presenting with MS-ON and undifferentiated ON, in comparison to the group with MS without ON, and the control population. At ON presentation, the

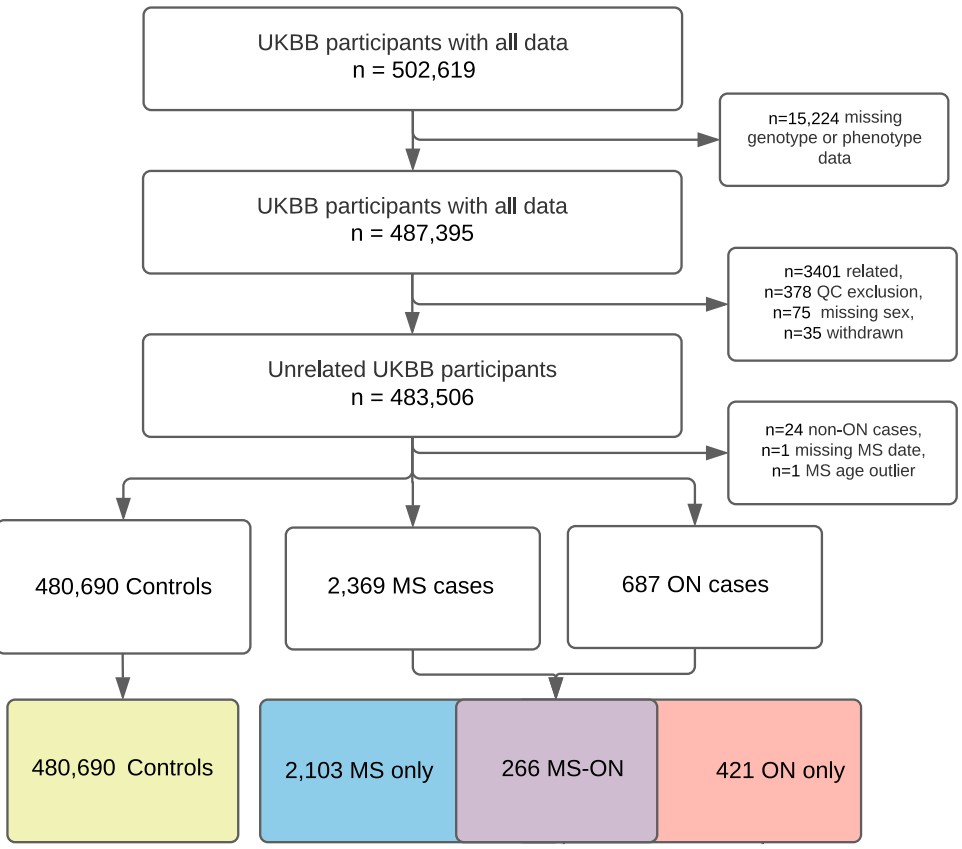

**Fig. 1 | Flow diagram of participants, illustrating exclusions and quality control steps.** The figure describes the exclusion and inclusion criteria used in the UK Biobank (UKBB) population. The boxes in the last row show the number of participants within each group: healthy controls (yellow), MS without ON (MS only, blue), MS-associated ON (MS-ON, purple), and ON without MS (ON only, red). An extended version is available in the supplement, which describes sources of diagnoses in more detail (Supplementary Fig. 1). UKBB UK Biobank, QC quality control, ON optic neuritis, MS multiple sclerosis.

**Table 1 | Demographic characteristics of participants presenting with MS-ON versus undifferentiated ON, and comparison with healthy control population, and group with MS without ON (MS Only) between three datasets**

| | | MS-ON at presentation | MS Only | Controls | Undifferentiated ON at presentation | Hazard ratio of MS diagnosis in undifferentiated ON |
|---|---|---|---|---|---|---|
| Study participants (n) | UKBB | 142 | 2103 | 480,690 | 545 | |
| | Geisinger | 280 | 1901 | 113,751 | 835 | |
| | FinnGen | 262 | 1544 | 369,633 | 977 | |
| Of which European ancestry (%) | UKBB | 121 (85.2) | 1845 (87.7) | 403,051 (83.9) | 462 (84.8) | 0.77 (0.48–1.24), P = 0.28 |
| | Geisinger | 247 (88.2) | 1656 (87.1) | 96,569 (84.9) | 751 (89.9) | |
| | FinnGen | NA | NA | NA | NA | |
| Median age at cohort enrolment (IQR, range), years | UKBB | 54.5 (48–59, 40 to 69) | 56 (49–62, 40 to 70) | 58 (50–63, 37 to 73) | 57 (50–62, 40–70) | 0.94 (0.92–0.97), P < 0.0001** |
| | Geisinger | 52.4 (45–62, 23 to 84) | 56 (49–62, 40 to 70) | 58 (50–63, 37 to 73) | 59.8 (48–73, 14–89) | |
| | FinnGen | 46.6 (37–55, 9 to 82) | 50.3 (40–60, 10 to 90) | 56.2 (41–67, 0.0 to 105) | 49 (37–62, 6–95) | |
| **n Females (F:M)** | UKBB | 106 (2.9) | 1504 (2.5) | 260,093 (1.2) | 352 (1.8) | **2.20 (1.41–3.45), P = 0.0005*** |
| | Geisinger | 220 (3.7) | 1440 (3.1) | 68,175 (2.4) | 546 (1.9) | |
| | FinnGen | 216 (4.7) | 1133 (2.8) | 206,268 (1.3) | 711 (2.7) | |
| **ON diagnosed between 18 and 50 years of age** | UKBB | 89 (62.3) | NA | NA | 336 (62.0) | **2.43 (1.43–4.17), P = 0.0014*** |
| | Geisinger | 125 (44.6) | NA | NA | 259 (31.0) | |
| | FinnGen | 220 (84.0) | NA | NA | 705 (72.2) | |
| **Mean MS-GRS (SD)** | UKBB | 3.74 (1.15) | 3.71 (1.17) | 2.66 (1.15) | 3.17 (1.30) | **1.29 (1.07–1.55), P = 0.0067*** |
| | Geisinger | 3.80 (1.28) | 3.35 (1.31) | 2.74 (1.14) | 2.92 (1.23) | |
| | FinnGen | 3.96 (1.24) | 3.73 (1.25) | 2.74 (1.13) | 3.41 (1.30) | |
| Variables not included in Cox regression | | | | | | |
| Mean age at onset ON (SD, range), years | UKBB | 47.3 (12.2, 20 to 73) | NA | NA | 44.7 (15.0, 1 to 80) | |
| | Geisinger | 40.8 (11.5, 15.7 to 69.6) | NA | NA | 52.2 (17.1, 11 to 89) | |
| | FinnGen | 38.7 (11.3, 10.5 to 72.9) | NA | NA | 38.8 (15.4, 7 to 89) | |
| Mean age at onset MS (SD, range), years | UKBB | 38.7 (9.6, 18 to 58) | 44.9 (12.5, 15 to 80) | NA | 45.5 (11.2, 20 to 73) | |
| | Geisinger | 45.0 (12.1, 16 to 77) | 47.1 (13.0, 4 to 88) | NA | 40.4 (11.9, 15 to 80) | |
| | FinnGen | 33.5 (9.7, 11 to 68) | 41.3 (12.6, 13 to 87) | NA | 36.3 (11.3, 15 to 76) | |
| Mean Non-HLA-GRS (SD) | UKBB | 2.86 (0.86) | 2.89 (0.93) | 2.37 (0.92) | 2.63 (0.98) | |
| | Geisinger | 3.12 (0.90) | 2.85 (0.98) | 2.50 (0.91) | 2.62 (0.95) | |
| | FinnGen | 3.11 (0.86) | 3.01 (0.93) | 2.44 (0.90) | 2.86 (0.99) | |
| Mean HLA-GRS (SD) | UKBB | 0.88 (0.74) | 0.73 (0.795) | 0.29 (0.71) | 0.53 (0.77) | |
| | Geisinger | 0.68 (0.82) | 0.50 (0.78) | 0.25 (0.69) | 0.30 (0.75) | |
| | FinnGen | 0.85 (0.85) | 0.72 (0.78) | 0.30 (0.69) | 0.55 (0.77) | |

P-values in the rightmost column are derived from a univariate MS-free Cox proportional hazard model in the undifferentiated ON UKBB population, unless specified otherwise with an asterisk: *P-values from multivariate Cox MS-free survival model with binary age at ON diagnosis, sex, and MS-GRS; **P-values from a model with binary age at ON diagnosis, sex, MS-GRS and age at UK Biobank (UKBB) enrolment.
Variables included in the final MS-free survival model are highlighted in bold.

female to male ratio was 1.8 in the initially undifferentiated ON group (n = 545) and 2.9 in the MS-ON group (n = 142), in comparison to 2.5 in the group with MS alone (n = 2103), and 1.2 in the control population (n = 480,690) ($\chi^2$, 9 degrees of freedom, P < 0.00001). The mean age at onset of undifferentiated ON was 44.7 (SD 15.0, range 1 to 80) years compared to 47.3 (12.2, range 20 to 73) years in the group presenting with MS-ON. The percentage of European ancestry cases was 84.8% (n = 462) in undifferentiated ON and 85.2% (n = 121) in MS-ON at presentation, compared to 87.7% (n = 1845) in the MS without ON group, and 83.9% (n = 403,051) in the control population. Supplementary Tables 6 and 7 present the data for all ON and all MS by end of cumulative follow-up.

### MS-GRS was discriminative of MS
We assessed the MS-GRS in MS cases and the rest of the UKBB (including ON without MS) (Fig. 2a). Both the HLA and non-HLA mean scores were higher in people with MS (mean 0.74 (SD 0.78) vs 0.29 (SD

0.71), P < 0.0001 for HLA, 2.88 (SD 0.92) vs 2.37 (0.92), P < 0.0001 for non-HLA) and were discriminative of MS (ROC-AUC (95% CI) 0.666 (0.663–0.669) and 0.656 (0.653–0.659) respectively, Fig. 2 and Supplementary Fig. 2). The full MS-GRS had a ROC AUC of 0.721 (0.718–0.723), and 0.752 (0.750–0.755) when combined with a subset of risk factors associated with MS (sex, age at UKBB entry, Townsend deprivation index) and the first four genetic principal components (Fig. 2a, d).

### Genetic overlap of MS, ON and MS-associated ON
We assessed the distribution of MS-GRS in healthy controls, ON only (undifferentiated ON at the end of follow-up), MS-ON and MS only (Fig. 2a–c), with groups defined by diagnosis at the end of cumulative follow-up. We found that the MS-GRS distribution of the non-MS-ON group (mean 3.02 (SD 1.29) lay between that of healthy controls 2.66 (1.15) and MS-ON cases 3.71 (1.17) (P < 0.0001 for both, Fig. 2a). The MS-GRS significantly differentiated MS cases from cases with non-MS-ON

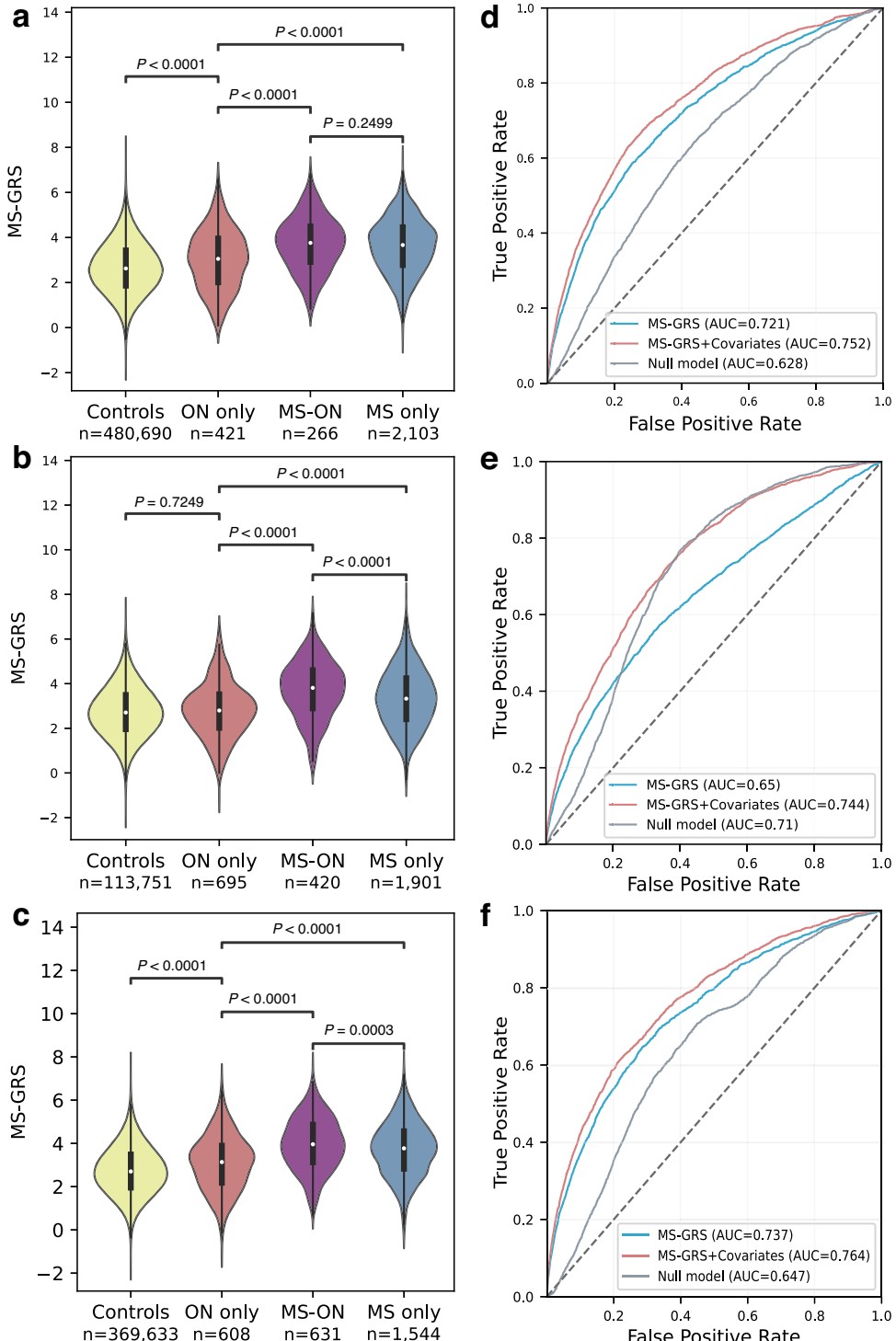

**Fig. 2 | MS-GRS distribution and ROC-AUC analysis across three cohorts.**
**a–c** MS-GRS distribution violin plots: comparative distribution of MS-GRS (multiple sclerosis genetic risk score) among different participant groups in three datasets: UK Biobank (**a**), Geisinger (**b**), and FinnGen (**c**). Groups are defined on the x-axis: healthy controls (Controls), individuals with optic neuritis without MS (ON only), MS-associated optic neuritis (MS-ON), and individuals with MS without optic neuritis (MS only). The mean is represented as a white circle, interquartile range as a black box, and the outside line shows the kernel density estimate of the underlying distribution. Each colour corresponds to a specific group: healthy controls (yellow), ON without MS (red), MS without ON (blue), and MS-ON (purple). The statistical analysis utilized two-sided Welch's t-test with Bonferroni correction term to account for multiple comparisons. **d–f** ROC-AUC analysis: receiver operating

characteristic area under the curve (ROC-AUC) analysis for differentiation between any form of MS (including MS only and MS-ON) versus healthy controls in three distinct datasets: UK Biobank (**d**), Geisinger (**e**), and FinnGen (**f**). The null model (grey line) encompassed the same covariates as the MS-GRS+covariates model (red line) but excluded the MS-GRS. MS-GRS without covariates is shown as a blue line. Covariates included in the models were: sex, TDI (Townsend Deprivation Index), age at cohort entry, and the first four principal components for UK Biobank; reported sex, index age, and the first four principal components for Geisinger; and sex, age at DNA sample collection, and the first four principal components for FinnGen. The ROC-AUC analysis provides insight into the discriminatory power of the models in distinguishing between MS cases and healthy controls.

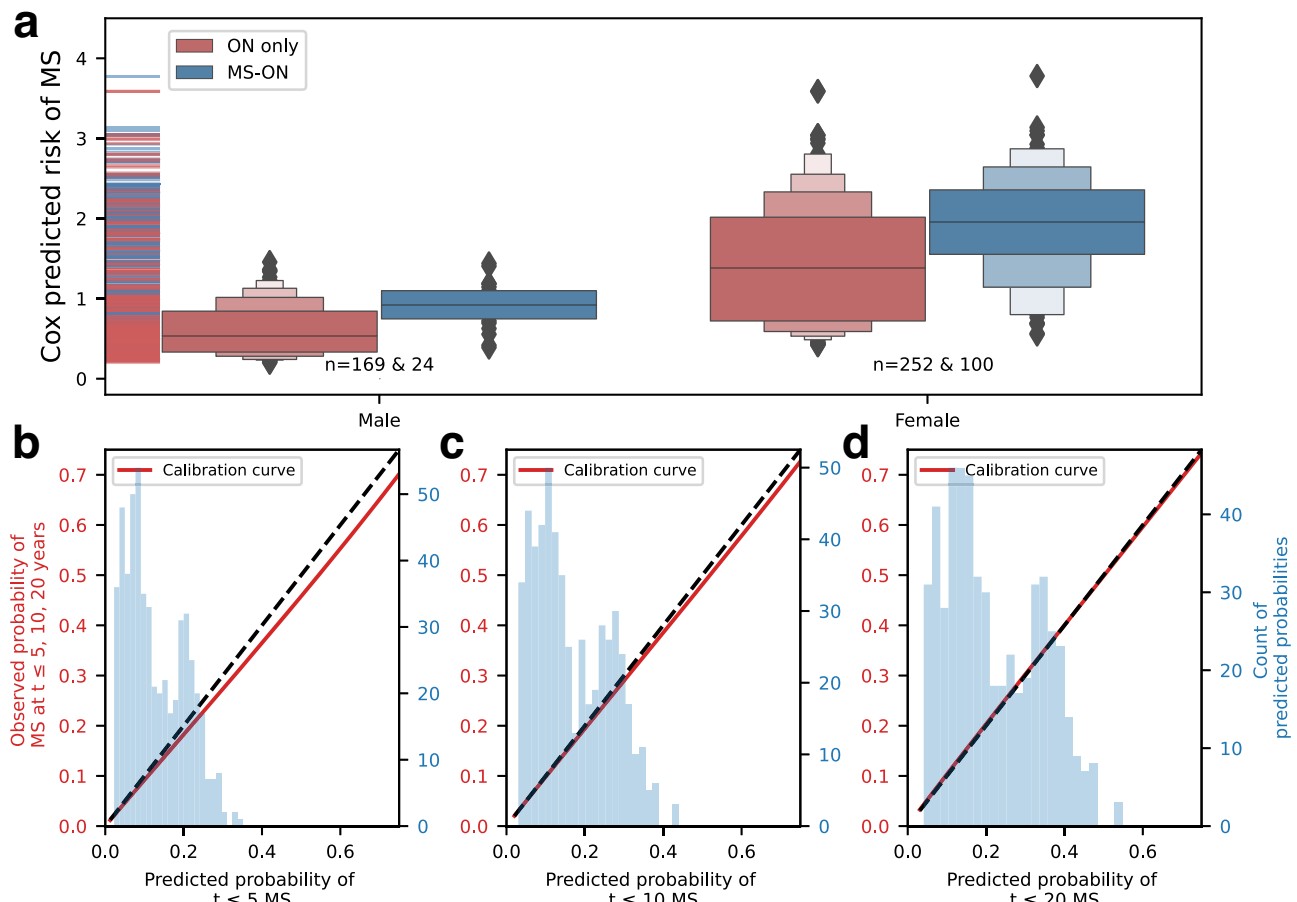

**Fig. 3 | Predicted risk of MS by Sex and model calibration plots for 5-, 10-, and 20-year horizons.** Panel (**a**) shows boxen plots of predicted partial hazard based on participants' MS-GRS, sex and age at ON diagnosis for undifferentiated ON that did not develop MS (ON only, red) and those who did (MS-ON, blue) by the end of cumulative follow-up. Grey centerline shows the median, with the darkest shade around showing second and third quartiles. Each successive level outward contains half of the remaining data and is shaded in lighter colour. Outliers are shown as diamonds. Panels (**b**–**d**) illustrate calibration plots of the Cox model at three points in time (5, 10, and 20 years, respectively). The smoothed calibration curve is shown in red, and ideal calibration as a black dotted line. X-axis is the predicted probability of developing MS up to 5, 10, and 20 years post ON diagnosis for (**b**–**d**), respectively.

at the end of study follow-up (3.62 (1.22), $P < 0.0001$), but not from cases with MS-ON (Table 1 and Fig. 2a, Supplementary Table 6). Figure 2 further demonstrates the distribution of MS-GRS in external validation datasets, with in-detail description provided in Supplementary Results. In individuals with MS-ON, MS-GRS did not differ significantly whether MS diagnosis preceded or followed first ON presentation in UKBB, although deviations from this were observed in one of the external datasets (Supplementary Fig. 9). There was a weak association between MS-GRS and age at onset of MS ($R^2 = 0.011$, $P < 0.0001$); higher MS-GRS was associated with younger age at MS diagnosis (Supplementary Fig. 10).

**MS-GRS predictive of future MS after first diagnosis of ON**

For the MS-free survival analysis, we limited our primary analysis of MS-GRS to undifferentiated ON cases ($n = 545$) after excluding people with MS diagnosed before ON ($n = 122$), and cases with first presentation of MS including ON ($n = 20$). This group had a median cumulative follow-up period of 18.4 (IQR 10–30) years. We included both prevalent ON (435/545 diagnosed before UKBB entry) and incident ON (110/545, diagnosed after UKBB entry). The outcome event, MS diagnosis, was documented in 22.9% ($n = 124$) cases, at a median interval of 3.8 years (IQR 0.8–12.2) years from ON to MS diagnosis.

Significant variables in Cox proportional hazard model single variable analysis included sex, binary age at ON diagnosis and MS-GRS only (Table 1 and Supplementary Table 5). These variables all remained

significant in multivariable analysis. Ancestry-associated principal components, and interaction between age at ON onset and sex were not significant in the multivariable model and were excluded (Supplementary Results 2.6). Proportional hazard assumptions were met in the final model at $P < 0.05$ in UKBB (Scaled Schoenfeld's residuals Supplementary Fig. 14). The model containing MS-GRS, sex and binary age at diagnosis calibrated well at 5, 10, and 20 years of follow-up (Fig. 3b–d) and the distribution of predicted cumulative MS risk (expressed as predicted partial hazard) is shown in Fig. 3a. MS-GRS was significantly associated with future development of MS, with adjusted Hazard Ratio (95% CI) of 1.29 (1.07–1.55), $P < 0.01$ per one standard deviation increase in MS-GRS. Stratification by quartiles of predicted risk (Fig. 4, Supplementary Fig. 15) identified individuals who, at different durations of follow-up, were at relatively low risk of MS (Percent diagnosed with MS at the end of the follow-up, 3.6%, 95% CI 0.5–6.8%), intermediate risk (14.7%, 8.8–20.7%), higher risk (31.6%, 23.8–39.4%) and highest MS risk (41.2%, 32.9–49.4%) (Fig. 5a). The median predicted partial hazard for each quartile is displayed in Supplementary Fig. 15. The sex-difference by quartile of predicted risk is illustrated in Fig. 5d and Supplementary Fig. 16. Lastly, we evaluated whether a full model (MS-GRS, binary age at onset, sex) performed better than covariates alone using time-dependant ROC-AUC up to 35 years (Supplementary Fig. 4), and found the average time-ROC-AUC for the full model was 0.627 vs 0.609 for the null model. It is worth mentioning that both absolute values and the difference between AUCs were

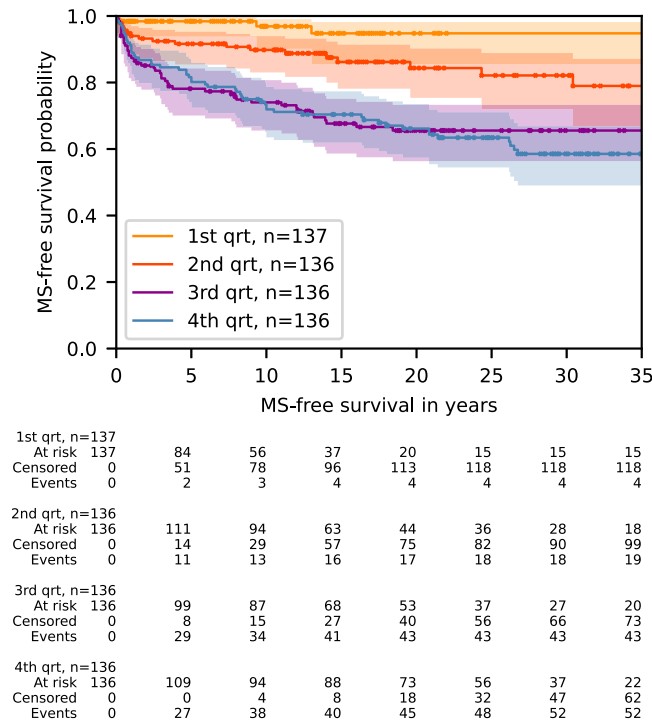

| 1st qrt, n=137 | | | | | | | |
| --- | --- | --- | --- | --- | --- | --- | --- |
| At risk | 137 | 84 | 56 | 37 | 20 | 15 | 15 | 15 |
| Censored | 0 | 51 | 78 | 96 | 113 | 118 | 118 | 118 |
| Events | 0 | 2 | 3 | 4 | 4 | 4 | 4 | 4 |

| 2nd qrt, n=136 | | | | | | | |
| --- | --- | --- | --- | --- | --- | --- | --- |
| At risk | 136 | 111 | 94 | 63 | 44 | 36 | 28 | 18 |
| Censored | 0 | 14 | 29 | 57 | 75 | 82 | 90 | 99 |
| Events | 0 | 11 | 13 | 16 | 17 | 18 | 18 | 19 |

| 3rd qrt, n=136 | | | | | | | |
| --- | --- | --- | --- | --- | --- | --- | --- |
| At risk | 136 | 99 | 87 | 68 | 53 | 37 | 27 | 20 |
| Censored | 0 | 8 | 15 | 27 | 40 | 56 | 66 | 73 |
| Events | 0 | 29 | 34 | 41 | 43 | 43 | 43 | 43 |

| 4th qrt, n=136 | | | | | | | |
| --- | --- | --- | --- | --- | --- | --- | --- |
| At risk | 136 | 109 | 94 | 88 | 73 | 56 | 37 | 22 |
| Censored | 0 | 0 | 4 | 8 | 18 | 32 | 47 | 62 |
| Events | 0 | 27 | 38 | 40 | 45 | 48 | 52 | 52 |

**Fig. 4 | Kaplan–Meier analysis of MS-free survival.** Kaplan–Meier analysis demonstrating MS-free survival trends based on quartiles of predicted MS risk. Quartile divisions were determined by the forecasted partial hazard of Multiple Sclerosis (MS), derived from individual characteristics utilizing the UK Biobank (UKBB)-trained Cox model. The time span to the event is calculated from the onset of optic neuritis (ON) diagnosis to the identification of MS, or the conclusion of follow-up for cases subjected to censoring. MS-free survival curves depicting quartiles of predicted risk for the validation cohort are provided in Supplementary Fig. 3 for further reference.

lowest in UKBB compared to other datasets, which could be explained by the lack of standardised follow-up in UKBB and overfitting of covariates.

### External validation

The MS prevalence was 1285 per 100,000 people in Geisinger and 556 per 100,000 people in FinnGen. In the FinnGen database, there were 977 cases of undifferentiated ON, of whom 369 (37.8%) developed MS with median of 1.02 years (IQR 0.24 to 5.46 years). In the Geisinger database, there were 835 cases of undifferentiated ON, of whom 140 (16.8%) developed MS with median latency of 0.32 years (IQR 0.06–1.68). MS-GRS was higher in both MS-ON and MS alone than either healthy controls or non-MS-ON in both validation cohorts (Fig. 2). It is worth mentioning that in both Geisinger and FinnGen MS-ON cases had higher MS-GRS than MS cases without ON—this was not observed in UKBB, and to our knowledge, has not been previously reported (Fig. 2b, c). MS-GRS was discriminative of MS cases vs healthy controls in both Geisinger (0.744 for MS-GRS with sex, index age, first four principal components, 0.650 for MS-GRS alone) and FinnGen (0.764 for MS-GRS with sex, age at DNA sample collection, and the first four principal components, 0.737 for MS-GRS alone) (Fig. 2e, f), and both HLA- and non-HLA were independently discriminative of MS (Supplementary Fig. 2). Amongst people with undifferentiated ON, the median risk of developing MS was 16.8% in the Geisinger population and 37.8% in FinnGen population. We used the multivariable Cox MS-free survival model trained on UKBB data in the Geisinger and FinnGen datasets and found that it calibrated well after adjusting for differing prevalence of MS in these different population cohorts (Supplementary Fig. 3). Using the model to split the data into quartiles of predicted

MS risk, we observed differing proportions of incident MS over cumulative follow-up of 5.8 years (IQR 1.6–10.8) in Geisinger, and 8.6 years (IQR 1.6–18.4) in FinnGen. Specifically, in the lowest quartile of MS-GRS predicted risk, 6.7% (95% CI 3.7–10.1%) in Geisinger, and 10.2% (95% CI 6.4–14.0%) in FinnGen developed MS. Whereas in the highest quartile of MS-GRS predicted risk, 30.6% (95% CI 24.4–36.9%) in Geisinger, and 60.7% (95% CI 54.5–66.8%) in FinnGen developed MS (Supplementary Fig. 3). Lastly, in Both Geisinger and FinnGen, full Cox survival model in undifferentiated ON (MS-GRS, sex, and binary ON diagnosis between 18 and 50) had better time-ROC-AUC for MS prediction than models containing covariates only (sex and binary age): 0.711 vs 0.692 in Geisinger, and 0.692 vs 0.647 in FinnGen (Supplementary Fig. 4).

### Sensitivity analyses

In subgroup analysis of European ancestry British individuals (84% participants), findings were similar to the main analysis (Supplementary Results Section 3). Specifically, the MS-GRS had a very similar ROC AUC of 0.750 (95% CI 0.746–0.753), when combined with risk factors associated with MS (sex, age at UKBB entry, Townsend deprivation index) and first four genetic principal components. Similarly, in the Cox proportional hazard model, the Hazard Ratio (95% CI) of future MS diagnosis amongst participants presenting with undifferentiated ON was 1.29 (1.05–1.58, *P* < 0.05) per standard deviation increase in MS-GRS.

In subgroup analysis of non-European ancestry British individuals (16% participants), findings were also similar to the main analysis (Supplementary Table 12), but without statistical significance, on account of the small number of participants. Specifically, the MS-GRS had a very similar ROC AUC of 0.753 (95% CI 0.746–0.760) when combined with risk factors associated with MS (sex, age at UKBB entry, Townsend deprivation index) and first four genetic principal components. Similarly, in the Cox proportional hazard model, the Hazard Ratio (HR) (95% CI) of future MS diagnosis amongst participants presenting with undifferentiated ON was 1.40 (0.89–2.22, *P* = 0.15) per standard deviation increase in MS-GRS versus 1.29 (1.07–1.55, *P* < 0.01) in the whole of UKBB. The HR for female sex was 1.81 (0.53–6.24, *P* = 0.35) versus 2.20 (1.41–3.45), *P* < 0.001), and the HR for age 18–50 years at ON diagnosis was 2.47 (0.69–8.77, *P* = 0.16) versus 2.43 (1.41–4.17, *P* < 0.001), as compared to the whole of UKBB, respectively.

Additional subgroup analyses restricted to either cases diagnosed after 20 years of age or diagnoses based on either hospital episode statistics (HES) or primary care records (GP records) revealed nearly identical results both for ROC-AUC MS-GRS performance and Cox models of future MS risk in undifferentiated ON. A summary of these subgroups and comparisons is provided in Supplementary Table 12.

### Pilot application

Here will illustrate how the combined model could be integrated into an application for use in clinical practice to estimate individual risk: https://mspredictor.com.

## Discussion

This pioneering investigation establishes a link between an individual's combined genetic susceptibility, as measured by the MS-GRS encompassing numerous MS-associated loci with common alleles, and the subsequent risk of MS development in those experiencing an initial episode of undifferentiated ON. Moreover, we unveil a stratification paradigm for individuals with undifferentiated ON, integrating the MS-GRS, age at ON onset, and sex, which delineates cohorts characterised by varying future MS risks: low (3.6%), intermediate (14.7%), higher (31.6%), and highest (41.2%). Significantly, our study demonstrates robustness through the successful replication and validation of the composite MS-GRS model across two distinct datasets from the USA and Finland, populations also predominantly of European ancestry.

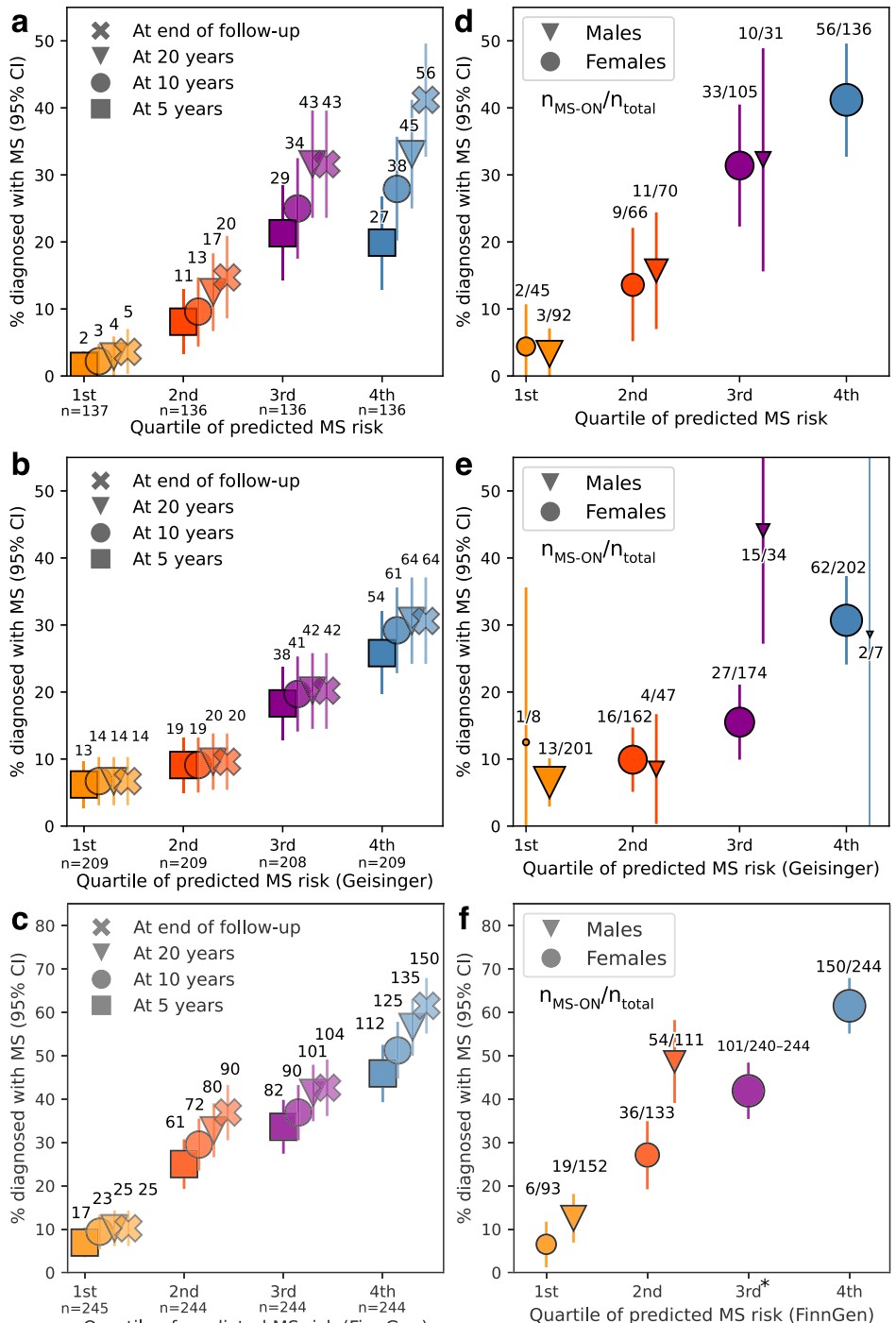

**Fig. 5 | Incidence of multiple sclerosis development in undifferentiated optic neuritis. a–c** Cumulative incidence graphs: Illustration of the percentage of participants experiencing undifferentiated optic neuritis (ON) within three distinct datasets: UK Biobank (**a**), Geisinger (**b**), and FinnGen (**c**). The analysis is stratified by quartile of predicted Multiple Sclerosis (MS) risk, with colours representing quartiles: orange—first, red—second, purple—third, and blue—fourth. The time points represent four horizons of cumulative follow-up, with squares representing 5 years, circles—10 years, triangles—20 years, and crosses—end of follow-up. Vertical lines denote 95% confidence intervals (95% CI) for the reported percentages estimated using normal approximation. Numbers adjacent to each marker indicate the cumulative number of people who developed MS in each quartile by time horizon. The total number of participants within each quartile are shown below the x-axis. **d–f** Sex-specific MS diagnosis rates: Graph depicting the percentage of each sex

who were subsequently diagnosed with MS by the end of follow-up, by quartile of predicted MS risk. Percentage calculated as number of females (circles) or males (triangles) with MS-ON divided by the total number of that sex within a quartile. The size ratio of markers within a quartile corresponds to the participant sex ratio. Numbers adjacent to each plot indicate the sex-specific number of participants with MS-ON versus the total number of either males or females in that group. Vertical lines indicate 95% CI for the sex-specific percentages estimated using normal approximation. Panel **d** is UK Biobank (UKBB), **e** is Geisinger, **f** is FinnGen cohort. "*" indicates a range, rather than exact value due to FinnGen's data protection policy on presenting potentially identifiable data. Remaining males (less than five participants), 100% of who developed MS, are not presented for this reason. It is important to note the differing y-axis scales in panels (**c**) and (**f**).

While it has been long-established that ON may be the first presentation of MS, the additional risk stratification outlined in this study could valuably aid management of ON, and greater international consensus on this[1], in the time-critical window before neuroimaging and serum and cerebrospinal fluid investigations are available. In usual clinical practice, European ancestry British women aged 18 to 50 years, who present with mild-moderate vision impairment, would not typically be offered corticosteroid therapy[8], and the MS-GRS would identify most of these individuals to be at enhanced MS genetic risk. Whereas, urgent corticosteroid therapy may be better targeted to the smaller number of individuals at low MS genetic risk, who are more likely to have an alternative, and potentially corticosteroid-responsive, cause for ON, averting irreversible vision loss. A low-risk MS-GRS could also reduce inadvertent initiation of interferon therapy in patients whose ON is later determined to be associated with NMOSD. Similarly, there may be value in avoidance of urgent corticosteroid therapy for a small number of patients aged less than 18 or over 50 years, whose presentation is 'atypical' for MS (e.g., vision worse than 6/60, no pain on eye movements or no improvement) but whose MS-GRS reveals high MS risk[15]. The availability of an MS-GRS combined model, in the context of undifferentiated ON, may help advance understanding of a clinically isolated syndrome, the forme fruste of ON in MS. Indeed, there is increasing recognition of MS clinical phenotypes falling on a continuum of disease severity and progression over time[16], and further research is needed to determine whether individuals at high MS risk should be directed to MS services more rapidly, for consideration of disease-modifying therapy.

## Limitations

The primary analysis was performed on individuals of all ancestries, with subgroup analyses on European and non-European ancestry British individuals. Ancestry was not a significant predictor of future MS diagnosis in patients presenting with undifferentiated ON, but non-European populations are underrepresented in UKBB. A recent study indicates that MS-GRS derived from predominantly white European ancestry populations do not translate well to South Asian ancestry populations[31]. It will be important to continue to develop large, diverse ancestry population biobanks and research studies to address this deficiency and avoid perpetuating health inequalities.

MS-free survival may have been affected by cohort intrinsic confounding, such as age at UKBB enrolment (Supplementary Fig. 18) and lack of standardised follow-up across the UKBB. Our Cox proportional hazard model included hazard ratios derived from the UKBB data. There is a risk of overfitting associations with sex and age at onset of ON because of the known selection biases in the UKBB data[32], including healthy volunteer bias. Our use of known risk factors from previous epidemiology and genome-wide association studies reduces the risk of a false result. However, we will in future test and calibrate our model using prospective diagnostic and implementation studies before a combined model can be integrated into clinical care. Additionally, UKBB data did not permit the use of the most recent and more precise HLA data[20] for HLA-GRS. Other general and important limitations of the UKBB study have been outlined elsewhere[33].

Classification of ON could be enhanced by using segmented optical coherence tomography (OCT) imaging data in the ON case definition[1]. Specifically, a 4%/4 μm inter-eye difference in the macula ganglion cell inner plexiform layer could enhance case definition, indicating prior unilateral optic nerve damage, as part of new diagnostic criteria for ON[34,35]. Unfortunately, OCT images are currently available for less than one-fifth of UKBB participants, and output from automated retinal image segmentation is not yet available in the public UKBB data repository to permit sensitivity analysis[36]. We anticipate that potential misclassification bias resulting from noise in case definitions, with likely overdiagnosis of optic neuritis based on diagnostic codes alone, would lead our findings to underestimate the value of MS-GRS in MS risk stratification, as compared to verification of ON cases with greater precision using OCT. Reassuringly, even though the UKBB is not population-representative and recruited adults aged over 40 years, we found comparable ON and MS prevalence to population-representative national studies[2,37].

Furthermore, while Epstein-Barr virus seropositivity was not available in the UKBB dataset, we explored diagnostic codes for prior clinical diagnosis of EBV infection or glandular fever, which have been identified as causal predictors of MS risk, but these diagnoses are scarcely available in UKBB[27]. Furthermore, seropositivity for EBV (data-field 23005) was available for less than ten thousand individuals. Finally, we would have liked to explore the additional contribution to future MS risk prediction of the presence or absence of brain lesions on unenhanced MRI imaging. However, this imaging was only performed in a subset of UKBB participants, and not at the time of ON diagnosis, and a variable relating to the presence or absence of demyelinating lesions suggestive of MS was not available for analysis.

## Rigorous quality control of phenotype data

Our study highlights that rigorous case definition QC in population biobanks is important and may reduce noise and increase power of analyses like ours. We manually checked all the diagnostic codes in each data source and performed a subgroup analysis which confirmed that similar results were found when cases were limited to those with a 'stricter' definition of MS and ON diagnosis. We also identified that two common Read3 codes for optic neuritis (F4H3 or F4H32) were omitted from the central UKBB definition of optic neuritis. This resulted in capture of an additional 194 cases of ON in this study. We also enhanced specificity for the diagnosis of ON by excluding a few codes used in the UKBB central ON definition, for example, 'optic neuropathy', which has many causes (e.g., genetic, nutritional, toxic, compressive) that are clinically distinct from ON. However, for 34 participants with undifferentiated ON who had only an ICD9 or ICD10 code, we were unable to further review which diagnostic codes made up the UKBB ON case definition.

## Comparison to existing literature

Our study leverages retrospective and prospective healthcare data to build on existing knowledge of the association between ON presenting as a clinically isolated syndrome (CIS) and future MS risk. The 1992 Optic Neuritis Treatment trial, which recruited 389 adults with acute unilateral, undifferentiated ON, reported a 5-year cumulative probability of MS of 29% rising to 38% at 10 years[7], and 50% by 15 years, with risk significantly associated with the presence of 1 or more lesions on baseline non-contrast-enhanced magnetic resonance imaging (MRI) of the brain[4], and also with female sex in participants without baseline MRI lesions (HR 3.6). This study found that the risk of developing MS was highest in the first 5 years following ON, and then decreased. A United States Armed Forces cohort study, including 1427 adults with ON, reported that 136 (9.5%) people developed MS by 10 years, including 19% of women and 14% of men with ON, and 68% were diagnosed within a year of ON[38].

In patients with undifferentiated ON, we found female sex to be a significant independent predictor of MS risk (aHR 2.20, $P < 0.005$, Fig. 3a), an association well-established in the literature but not fully understood, reflecting a complex interplay between genetic, epigenetic, immunological, hormonal and environmental factors[39,40]. Binary age at onset of ON (between 18 and 50 years) was also significantly associated with MS risk (aHR 2.43, $P < 0.005$). We found a weak inverse association between MS-GRS and age at MS onset ($R^2 = 0.011$, $P < 0.0001$, Supplementary Fig. 10), aligning with a recent study by Misicka et al. (2022) reporting higher MS-GRS risk burden and younger age at MS diagnosis[41]. We did not find significant associations with additional clinical risk factors including BMI, smoking, or vitamin D insufficiency at UKBB study entry, which have been highlighted in

other studies[22–25,28,29]. It is possible that this is because our study only measures these variables at a single time point (UK Biobank study entry) unrelated to diagnosis of ON and/or MS. Additionally, risk factors that are significantly associated with disease in large observational epidemiology studies sometimes do not explain enough variation in disease development to be useful for clinical prediction.

### Future research

Further research is needed to test the hypothesis that an MS-GRS combined with existing diagnostic[6], demographic, and other deep phenotypic variables, can usefully stratify patients with undifferentiated ON into high/medium/low genetic MS risk in a prospective diagnostic predictive clinical setting. Patient and public involvement around the acceptability of integrating genetic risk stratification into frontline care will be vital. Our study hints at the possibility of clinical translation, with use of a genetic test at first ON presentation to deliver better acute clinical care. With up to 5 million adults in the UK soon to be recruited into the UK's largest ever health research programme, including genomic medicine, 'Our Future Health', use of GRSs could soon become part of an enhanced approach to personalised medicine[42].

In summary, our study unveils the potential of a composite model that integrates MS-GRS with age at ON onset and sex, offering a means to stratify patients based on their likelihood of a future MS diagnosis, thus providing valuable insights for clinical management decisions. Future research endeavours should delve into the practical application of the MS-GRS model within clinical settings. We hypothesize that the knowledge of high MS-GRS, especially in individuals with suspected clinically isolated syndrome, would facilitate MS follow-up management and guide decisions around performing lumbar puncture to seek earlier MS diagnosis and potentially earlier disease-modifying treatment to reduce relapse rate. Additionally, we posit that a swift MS-GRS model panel test to identify low MS-GRS could facilitate hyperacute corticosteroid treatment, especially of subsequent vision-threatening ON relapses, potentially mitigating visual morbidity in those with non-MS-ON, while also yielding substantial economic and quality-adjusted life year benefits.

## Methods

### Data source and population

We studied participant data in the UKBB, a longitudinal population-based cohort study, described in detail elsewhere[43]. In brief, the UKBB comprises extensive genetic and phenotypic data from ~500,000 individuals ($n = 229,134$ men, $n = 273,402$ women) aged 40–69 years. Participants were recruited from 22 assessment centres in the UK between 2006 and 2010, with data linkage to hospital episode statistics (HES), the death register, and primary care data. UKBB participants gave informed consent to participate, and ethics committee approval was granted by the Northwest Multi-Centre Research Ethics Committee (ref 06/MRE08/65).

### Identifying cases and controls

Our case identification process is detailed in Supplementary Methods 1.1 and Supplementary Fig. 1, with a summary in Fig. 1. We define the included diagnostic codes in Supplementary Tables 1–3 and excluded 24 cases who had Read codes indicating a diagnosis distinct from ON, or to be insufficiently specific for ON diagnosis. We excluded cases with MS diagnosis before the age of 15.0 years, but retained cases with ON before the age of 15.0 years.

We defined four groups: MS without ON, MS with ON, ON without MS, and controls who had neither ON nor MS. All groups contained both prevalent and incident cases. We analysed the order of diagnoses using the earliest available date of diagnosis from all data sources (Hospital Episode Statistics (HES), GP records, self-report). We limited our MS-free survival analysis to undifferentiated ON cases, after excluding those with MS diagnosis preceding ON diagnosis, and compared those who were subsequently diagnosed with MS to those who were not.

### Genetic data

We used imputed genetic data downloaded from the UK Biobank[44]. We limited our analysis to 11,977,111 genetic variants imputed using the Haplotype Reference Consortium imputation reference panel with a minimum minor allele frequency (MAF) > 0.1% and imputation quality score (INFO) > 0.3. We used eight HLA alleles imputed to four-digit resolution centrally by UKBB using HLA*IMP:02[45].

The primary analysis was based on unrelated individuals of all ancestries. We excluded one of each pair of related individuals at random based on the genetic relatedness coefficient (>=0.084) to exclude third-degree relatives or closer, and to reduce the risk of bias from cryptic relatedness ($n = 3400$ in total, $n = 21$ with MS and $n = 4$ with ON)[46]. We performed a secondary analysis of individuals identified as European ancestry British by principle component analysis (Data-Field 22006), and then separately those of non-white-European ancestry.

### Generating the MS genetic risk score (MS-GRS)

We used external sources of risk alleles and odds ratios for non-HLA and HLA alleles (Supplementary Methods 1.2). Specifically, we generated a non-HLA genetic risk score using 317 autosomal single nucleotide polymorphisms (SNPs) with a genome-wide significance of $P$ value $< 10^{-5}$, including 200 SNPs with genome-wide significance $P < 5 \times 10^{-8}$ and a further 117 strongly suggestive SNPs with genome-wide significance between $P < 10^{-5}$ and $P > 5 \times 10^{-8}$ [20]. All SNPs were outside the extended HLA region (i.e. excluding the chromosome 6 region from 24 to 35 Mbps, hg19). We ensured that no SNPs were in linkage disequilibrium ($r^2 > 0.2$) using LDlink ($n = 8$)[47], and excluded ambiguous ($n = 1$), missing ($n = 1$), or duplicated ($n = 1$) variants, resulting in 307 SNPs (Supplementary Data 1). We calculated a log-additive sum of the risk alleles in PLINK2, using a natural log of odds ratios (log OR) as weights.

Recent work has revealed that accounting for HLA interaction improves the discriminative performance of autoimmune disease GRS[48,49]. A recent GWAS by IMSGC described HLA and nearby non-HLA genes in-detail, including independent effects within some loci[20]. However, we used a 10-allele HLA interaction model developed by Moutsianas et al. (2015) on 17,456 MS cases and 30,385 controls from across 11 cohorts to account for non-additive interaction between the HLA alleles derived externally from UKBB[50]. This included 8 imputed HLA alleles and 2 SNPs from the HLA region (29.9 to 33.6 Mbps on chr6 hg19). We captured interactions between the alleles by calculating the interactive model; scoring imputed HLA alleles while employing both additive effects, homozygote correction terms, and conditional scoring of some HLA alleles (Supplementary Table 4). We performed this scoring using the Python 3 libraries Pandas and NumPy[51,52]. We then scored the two SNPs from the HLA region by multiplying the natural log odds ratios (OR) by risk allele dosage using PLINK2. Lastly, we combined the scores calculated from HLA alleles and the two SNPs to produce the HLA-GRS. The final MS-GRS was a sum of non-HLA- and HLA-GRS.

### Statistical analysis

We analysed the distribution of MS-GRS in the four groups: healthy controls, MS only cases, ON only cases, and cases with both MS and ON, using Welch's t-test. We tested the ability of MS-GRS to discriminate between MS cases and healthy controls using the area under the curve (AUC) of the receiver operating characteristic (ROC). We compared the discriminative power of covariates only, MS-GRS only, and MS-GRS plus covariates. To avoid overfitting, we calculated each ROC-AUC using three-fold cross-validation with ten repetitions,

reporting means with 95% confidence intervals. Covariates selected from the published literature included sex, age at UKBB entry, Townsend deprivation index (TDI) and the first four genetic principal components, as they were previously scrutinised in the context of an MS genetic risk score[53].

Finally, to assess the ability of MS-GRS to predict MS-free survival time in cases presenting with undifferentiated ON (i.e., without prior MS diagnosis), we explored Cox proportional hazards survival regression, by multiple potential predictor variables specified a priori from literature review. Here, we tested potential predictor variables more extensively, including those putatively associated with MS. Binary variables included sex, age group (18–50 years versus younger and older), and ethnicity (European ancestry British versus not/unknown, Data-Field 22006)[22]. Continuous variables included age, MS-GRS (standardised for 545 individuals with undifferentiated ON), and Body Mass Index (kg/m², UK Biobank Data-Field 21001)[22,24–26]. Categorical variables included smoking status (ever vs never vs missing, Data-Field 20160)[28,29,54], country of birth (England, Scotland, Northern Ireland, Republic of Ireland, Wales and Other/Unknown, Data-Field 1647), Townsend deprivation index quintiles (1 to 5 or missing, Data-Field 189), and serum 25-OH vitamin D level at UKBB baseline assessment (sufficient [>50 nmol/L], insufficient [25–50 nmol/L], deficient [<25 nmol/L] or missing, Data-Fields 30890–30896)[22–24]. We assumed the age of diagnosis was the earliest record of diagnosis across all sources. For the outcome variable, we estimated the time from diagnosis of ON to diagnosis of MS using the earliest records of diagnoses available for both diseases. Censoring was estimated using the latest HES or GP episode record available for each individual, and where it was deemed unsuitable or was not available (e.g., last record preceded enrolment date, or neither HES nor GP records were available for an individual ($n = 26$ people with non-MS-ON) we used the last date of global HES update. Variables reaching statistical significance $P < 0.05$ in the single variable analysis were included in the full multivariable regression model and were removed through backward elimination to identify the most parsimonious model with the lowest partial AIC (Akaike information criterion). We considered the interaction term between sex and age at ON diagnosis.

We explored the impact of genetic stratification on our results in all UKBB participants by performing an analysis of the European ancestry population only, as defined by UKBB self-reported ethnicity and genetic principal components (Data-Field 22006) (Supplementary Results Section 3). While anticipating the analysis to be underpowered, we also performed a sensitivity analysis comparing non-European and European ancestry British participants. Two additional subgroup analyses included one on 'strict' diagnoses from either GP records or HES only, and one excluding cases diagnosed before 20 years of age (Supplementary Table 12).

Statistical analyses and visualisations were performed using Python 3 and NumPy[52], Scikit-learn[55], Matplotlib[56] and Lifelines libraries[57]. All codes for the completed analyses are available at https://github.com/ploginovic/MS-ON-ukb-code.

### External validation
We sought to validate our findings in two large genetic and health datasets, Geisinger, USA[58,59] and FinnGen, Finland[60,61]. We assessed discrimination and calibration of the UKBB combined model in these datasets. See Supplementary Methods and Results (Sections 1.3–2.1) for additional detail.

### Reporting summary
Further information on research design is available in the Nature Portfolio Reporting Summary linked to this article.

## Data availability
The Genetic risk score will be deposited in the Polygenic Score Catalog (PGS Catalog: https://www.pgscatalog.org/) upon receiving a DOI of this study. Individual-level genotype data described in this study are available to bona fide researchers as per the UK Biobank data-access protocol (https://www.ukbiobank.ac.uk/enable-your-research/apply-for-access). Further details and instructions about registration for access to the data are available at http://www.ukbiobank.ac.uk/register-apply/. UK Biobank accession codes of this study were 9055 and 9072. For FinnGen data, access to individual-level sensitive health data must be approved by national authorities for specific research projects and for specifically listed and approved researchers in accordance with the National and European regulations (GDPR). Researchers can apply for the health register data from the Finnish Data Authority Findata (https://findata.fi/en/permits/) and for individual-level genotype data from Finnish biobanks via the Fingenious portal (https://site.fingenious.fi/en/) hosted by the Finnish Biobank Cooperative FINBB (https://finbb.fi/en/). For Geisinger, the data was generated as described in Carey et al.[58]. Further details regarding phenotype and genotyping data for Geisinger can be found here: https://www.geisinger.org/precision-health/mycode/discovehr-project. Institutional Review Board determined this study to be "Non-human subject research" using de-identified information (IRB #: 2023-1075). The HLA genotyping data and MS-GRS from the MyCode participants in this study may be shared with a third party bona fide researchers upon execution of the data-sharing agreement.

## Code availability
The code used for phenotype, genotype, and statistical analysis is available through the following GitHub repository: https://github.com/ploginovic/MS-ON-ukb-code. Statistical analyses were performed in Python v3.10 (https://docs.python.org/release/3.10.11/), with adaptations the LifeLines Python package, covered by the MIT license (https://github.com/CamDavidsonPilon/lifelines). Genetic analyses were performed in PLINK v1.9 (https://www.cog-genomics.org/plink/) and PLINK v2.0.a (https://www.cog-genomics.org/plink/2.0/). Phenotype analyses were performed in STATA v17 (https://www.stata.com) and R v3.6 (https://www.r-project.org).

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

## Acknowledgements

T.B. was supported by the Royal College of Ophthalmologists and Fight for Sight Zakarian Award 2022 (RCOZAK2022). P.L. was supported by the INSPIRE Studentship by the University of Exeter. RAO had a UK MRC confidence in concept award to develop a type 1 diabetes GRS biochip with Randox R&D and has ongoing research funding from Randox; and has research funding from a Diabetes UK Harry Keen Fellowship (16/0005529), National Institute of Diabetes and Digestive and Kidney Diseases grants (NIH R01 DK121843–01 and U01DK127382–01), JDRF (3-SRA-2019–827-S-B, 2-SRA-2022–1261-S-B, 2-SRA-2002–1259-S-B, 3-SRA-2022–1241-S-B, and 2-SRA-2022–1258-M-B), and The Larry M and Leona B Helmsley Charitable Trust. This study was supported by the National Institute for Health and Care Research Exeter Biomedical Research Centre. The views expressed are those of the author(s) and not necessarily those of the NIHR or the Department of Health and Social Care. This research has been conducted using data from UK Biobank (https://www.ukbiobank.ac.uk/), a major biomedical database, and the authors are grateful to the participants. The authors are grateful to the participants of the Geisinger MyCode Community Health Initiative for the use of their genomic and electronic health information, without whom this study would not be possible. The patient enrolment and exome sequencing for the DiscovEHR study were funded by the Regeneron Genetics Center. We would like to acknowledge the Geisinger-Regeneron DiscovEHR Collaboration for making the genotype data and phenotype available for this project. We want to acknowledge the participants and investigators of FinnGen study. The FinnGen project is funded by two grants from Business Finland (HUS 4685/31/2016 and UH 4386/31/2016) and the following industry partners: AbbVie Inc., AstraZeneca UK Ltd, Biogen MA Inc., Bristol Myers Squibb (and Celgene Corporation & Celgene International II Sàrl), Genentech Inc., Merck Sharp & Dohme LCC, Pfizer Inc., GlaxoSmithKline Intellectual Property Development Ltd., Sanofi US Services Inc., Maze Therapeutics Inc., Janssen Biotech Inc, Novartis AG, and Boehringer Ingelheim International GmbH. The following biobanks are acknowledged for delivering biobank samples to FinnGen: Auria Biobank (www.auria.fi/biopankki), THL Biobank (www.thl.fi/biobank), Helsinki Biobank (www.helsinginbiopankki.fi), Biobank Borealis of Northern Finland (https://www.ppshp.fi/Tutkimus-ja-opetus/Biopankki/Pages/Biobank-Borealis-briefly-in-English.aspx), Finnish Clinical Biobank Tampere (www.tays.fi/en-US/Research_and_development/Finnish_Clinical_Biobank_Tampere), Biobank of Eastern Finland (www.ita-suomenbiopankki.fi/en), Central Finland Biobank (www.ksshp.fi/fi-FI/Potilaalle/Biopankki), Finnish Red Cross Blood Service Biobank (www.veripalvelu.fi/verenluovutus/biopankkitoiminta), Terveystalo Biobank (www.terveystalo.com/fi/Yritystietoa/Terveystalo-Biopankki/Biopankki/) and Arctic Biobank (https://www.oulu.fi/en/university/faculties-and-units/faculty-medicine/northern-finland-birth-cohorts-and-arctic-biobank). All Finnish Biobanks are members of BBMRI.fi infrastructure (www.bbmri.fi). Finnish Biobank Cooperative -FINBB (https://finbb.fi/) is the coordinator of BBMRI-ERIC operations in Finland. The Finnish biobank data can be accessed through the Fingenious® services (https://site.fingenious.fi/en/) managed by FINBB.

## Author contributions

Conceptualization: R.A.O., T.B., L.F. and P.L. Genetic analysis: P.L., F.W., J.L., L.F., M.N.W., H.S.R. and R.A.O. Phenotype preparation, harmonization, and analysis: P.L., F.W., J.L., T.B., L.F., M.N.W., J.T., H.D.G., U.L.M., H.S.R., D.J.C. and R.A.O. Statistical analysis and modelling: P.L., F.W., J.L., L.F., T.B., R.A.O., H.D.G., M.N.W. and U.L.M. Project administration: R.A.O., T.B., M.N.W., T.T., A. G. and D.J.C. Supervision: T.B., R.A.O., T.T., D.J.C. and A.G. Writing—original draft: T.B., R.A.O., P.L, L.F., A.P., M.N.W. and J.T. Data curation: M.N.W., U.L.M., D.J.C., T.T. and A.G. All authors contributed to data interpretation, manuscript revisions, and approval of the final version of the manuscript.

## Competing interests

R.A.O. is a co-investigator on a Randox Laboratories R&D research grant and received translational industry academic funding from Randox Laboratories R&D relating to autoimmune GRS for prediction and classification of disease. There are no established patents, loyalties, or licensing agreements relating to this grant. It is a 3-year grant (February 2022–2025). The approximate value is a £2.2 million program grant on GRS across autoimmune disease. A.P. reports personal fees from Novartis, Heidelberg Engineering, Zeiss, grants from Novartis, outside the submitted work; and is part of the steering committee of the OCTiMS study which is sponsored by Novartis and the Angio-OCT steering committee which is sponsored by Zeiss. He does not receive compensation for these activities. Other authors have no competing interests to declare.

## Additional information

[1]University of Exeter Medical School, College of Medicine and Health, University of Exeter, Heavitree Road, Exeter EX1 2HZ, UK. [2]Institute for Molecular Medicine Finland (FIMM), HiLIFE, University of Helsinki, Helsinki, Finland. [3]Weis Center for Research, Geisinger, Danville, PA, USA. [4]Institute of Biomedical and Clinical Science, University of Exeter Medical School, St Luke's Campus, University of Exeter, Heavitree Road, Exeter, Devon EX1 2LU, UK. [5]Neuro-ophthalmology Expert Center, Amsterdam UMC, Amsterdam, The Netherlands. [6]Department of Neuro-ophthalmology, The National Hospital for Neurology and Neurosurgery, Queen Square, UCL Institute of Neurology, London, UK. [7]Neuro-ophthalmology service, Moorfields Eye Hospital, London, UK. [8]Genetics of Complex Traits, University of Exeter Medical School, University of Exeter, Exeter EX2 5DW, UK. [9]Exeter Centre of Excellence for Diabetes Research (EXCEED), University of Exeter Medical School, St Luke's Campus, University of Exeter, Heavitree Road, Exeter, Devon EX1 2LU, UK. [10]Analytic and Translational Genetics Unit, Department of Medicine, Massachusetts General Hospital, Boston, MA, USA. [11]Abdominal Center, Endocrinology, University of Helsinki and Helsinki University Hospital, Helsinki, Finland. [12]Folkhälsan Institute of Genetics, Folkhälsan Research Center, Biomedicum, Helsinki, Finland. [13]Lund University Diabetes Centre, Department of Clinical Sciences, Lund University, Malmö, Sweden. [14]Academic Kidney Unit, Royal Devon University Healthcare NHS Foundation Trust, Exeter, UK. [15]King's College London, School of Immunology & Microbial Sciences and School of Life Course and Population Sciences, London, UK. [16]Medical Eye Unit, St Thomas' Hospital, Guy's and St Thomas' NHS Foundation Trust, Westminster Bridge Road, London, UK. [17]These authors contributed equally: Pavel Loginovic, Feiyi Wang, Jiang Li. [18]These authors jointly supervised this work: Richard A. Oram, Tasanee Braithwaite. ✉e-mail: R.Oram@exeter.ac.uk

## UKBB Eye & Vision Consortium

**Tasanee Braithwaite** [15,16,18], **Richard A. Oram** [4,14,18] ✉, **Axel Petzold** [5,6,7] **& Michael N. Weedon**[4]

## FinnGen

**Tiinamaija Tuomi** [2,11,12,13] **& Andrea Ganna**[2,10]

## Geisinger-Regeneron DiscovEHR Collaboration

**David J. Carey**[3]

