## [Peer Review File · Nature Communications]

Applying a genetic risk score model to enhance prediction of future multiple sclerosis diagnosis at first presentation with optic neuritisEditorial Note: This manuscript has been previously reviewed at another journal that is not operating a transparent peer review scheme. This document only contains reviewer comments and rebuttal letters for versions considered at *Nature Communications* .

REVIEWERS' COMMENTS

Reviewer #1 (Remarks to the Author):

The revised manuscript is a meaningful improvement, allowing the reader to transparently evaluate the comprehensive suite of analyses that were well presented.

Minor comments:

Line 18: Please remove "excellently" as the sentence lacks qualifiers.

Figure 1. Why is the MS N=2369 lower than in eFigure 1?

eFigure1. There are 2383 MS cases, but it only ends up with 2117 with MS only and 266 with MS-ON – 1 MS case is not classified. Is this correct?

Line 107 and Supplement 1.2 Line 199. Do acknowledge are only the autosomal risk variants

Nature Communications Peer Review Feedback
Received 21 November 2023

Article entitled "Applying a genetic risk score model to enhance prediction of future Multiple Sclerosis diagnosis at first presentation with optic neuritis"

Reviewer	Remarks to Author	Our response
1	The revised manuscript is a meaningful improvement, allowing the reader to transparently evaluate the comprehensive suite of analyses that were well presented.	Thank you very much.
1	Line 18: Please remove "excellently" as the sentence lacks qualifiers.	Thank you for your suggestion. We have removed the word "excellently" from the abstract in the process of shortening it.
1	Figure 1. Why is the MS N=2369 lower than in eFigure 1?	Thank you very much for pointing this out. An old version of the figure has been provided in error. The provided figure was created early in the process before removing participants based on relatedness and integrating icd9 code for MS. Hence, the numbers in the supplementary Venn diagram did not add up to the final number of MS cases provided in Fig. 1. We have updated eFigure 1 with an up-to-date flow diagram and explained exclusion reasons more clearly.
1	eFigure1. There are 2383 MS cases, but it only ends up with 2117 with MS only and 266 with MS-ON – 1 MS case is not classified. Is this correct?	In the old figure, it was not explained that one of the people with MS was removed as they also were 1 of the 24 individuals removed due to having a diagnosis of nutritional or drug-induced optic neuropathy. We now included this in eFigure 1 as an exclusion reason. Thank you for pointing this out.
1	Line 107 and Supplement 1.2 Line 199. Do acknowledge are only the autosomal risk variants	Thank you for your suggestion. We added "autosomal" in the following sentence on line 411 (page 14, paragraph 2 (Previously line 108, page 4, para 1)): "Specifically, we generated a non-HLA genetic risk score using 317 autosomal single nucleotide polymorphisms (SNPs)..." In Supplementary Methods 1.2 (Line 110, para 1, page 7), we added "First, we extracted 317 autosomal variants associated with MS from summary statistics of IMSCG GWAS meta-analysis discovery phase.³". In Supplement 1.2 (Line 114-115), we also state that we excluded sex-chromosome variants: "We excluded all variants from the extended HLA region on chromosome 6 (chr6:25 Mbp to chr6:35 Mbp, hg19), and sex-chromosome variants."